# Developing Personas of Gamers with Problematic Gaming Behavior among College Students Based on Qualitative Data of Gaming Motives and Push–Pull–Mooring

**DOI:** 10.3390/ijerph20010798

**Published:** 2023-01-01

**Authors:** Shan-Mei Chang, Sunny S. J. Lin

**Affiliations:** 1School of Nursing, China Medical University, Taichung 406040, Taiwan; 2Nursing Department, China Medical University Hospital, Taichung 404327, Taiwan; 3Tsing Hua Interdisciplinary Program, National Tsing Hua University, Hsinchu 30044, Taiwan; 4Institute of Education, National Yang Ming Chiao Tung University, Hsinchu 30010, Taiwan

**Keywords:** gaming motives, problematic gaming behavior, college student, content analysis, persona, push–pull–mooring

## Abstract

Gaming is a popular but possibly problematic activity among college students. To distinguish gamers with potential problematic gaming behaviors (PGB) is crucial to mental health staff. Two studies were conducted that aimed to explore portraits of gamers with PGB in college campuses. The first study selected 20 college students, diagnosed with problematic gaming behaviors, from a longitudinal dataset and semi-structured interviews were conducted for a systematic description of long-term PGB. The second study selected four personas with the richest coding data of internet addiction and depression from 20 gamers. The profiles and life experiences of the personas showed changing processes of gaming motives and push–pull–mooring effects across the years. “Loss of purpose in life” and “desperate to escape from stress or boredom in the real world” were the important push effects. Mooring effects revealed their addiction or depression symptoms and the process of developing the addiction. The dynamics of “push”, “pull”, and “mooring” effects were clearly indicated in the results suggesting PGB might be a long-term coping strategy and a consequence of depression and loneliness. Dealing with depression and finding real-life goals could help PGB gamers to change the dynamics of their gaming motives and push–pull–mooring effects. The results may help develop interventions for gamers with problematic gaming behaviors.

## 1. Introduction

In the past two decades, online games have become one of the most important online activities and social platforms among college students [1,2,3]. Their popularity can be demonstrated through many examples. For example, in Taiwan 65% of all online users have played online games (or mobile games) at least once a month [4]. In 2018, gaming disorder was officially included in the non-substance addictions in psychiatric diagnosis [5]. Then in 2020, the outbreak of COVID-19 has forced the changing of many activities from offline to online [6,7,8], resulting in the fact that many gamers spent more time playing online games or surfing online during lockdown than before [9]. Nonetheless, research [10] indicated that only a small proportion, 3%, of adolescent gamers (*n* = 8110) were diagnosed as having a gaming disorder. To provide the portraits and life experiences of gamers with PGB could be helpful for identification and diagnosis.

Online games involve many people through cooperation and social interactions [1] and the motives of gamers are worthy of investigation [11,12,13]. On the one hand, the rewards of online games are delicately designed to have extreme attraction [14,15] which is the extrinsic component regarding gamers’ motivation. On the other hand, the achievement gained from upgrades in games [16] or interactions with other gamers may be the intrinsic component of a player’s psychological needs [17]. Online games were defined as an “active” leisure activity and many experience fun [18]. Feeling fun could be the reason why gamers engage and immerse heavily in the game world [11]. In addition, previous studies [19] have suggested that problematic gaming is likely to result from a gamer’s use of online games as a strategy to cope with their real live stress. Identifying gamers’ motives may be helpful in finding out which gamers have a problem [12,15,20] as previous studies have found three gaming motives related to problematic gaming behaviors: escapism [15,18,20,21], achievement [20,22,23], and socializing [24,25,26].

The well-designed fun-seeking mechanisms could invite gamers to stay in games and thus may lead gamers to spend excess time or resources (such as money) than they can afford. Therefore, Hou et al. [27] applied push–pull–mooring (PPM) theory to Massively Multiplayer Online Role-Playing Game (MMORPG) gamers to explain the over immersion observed among different games. Another study [2] applied the PPM framework to demonstrate the pulls from the virtual world leading the gamers to transition among different games, one after another. Not only do gamers switch among different MMORPGs [27], but also bloggers switch among different blog platforms [28,29]. These studies mostly focused on migration within games of the online world. The original PPM model describes “push” as a negative factor, such as natural disasters or unemployment, which lead people to move out from a region. On the contrary, “pull” is a positive factor, such as low living costs, making a new region attractive to move to. In addition, Longino and Serow [30] added the term of “moor” to explain how personal factors (such as lifestyle) are related to migration decisions. Chang and Lin [20] adopted the PPM model to interpret how and why gamers transit between the virtual world and the real world. They suggested that there are push forces to drag an individual from the real world into the virtual world; while pull is the force that entices an individual from the real world to the virtual world. They further used three gamming motives (socializing, escapism, and advancement) to demonstrate push and pull effects.

Most problematic gaming studies are conducted with self-reported quantitative surveys [22,23,31] in order to synthesize and simplify common features or structural relations of risk behaviors. Only a few qualitative studies [19] have been conducted in selected/identified gamers with problematic gaming behavior. Many anecdotes or un-proven comments were dispatched related to gaming disorder, but gaming disorder is a relatively new medical diagnosis with precise diagnostic criteria [5]. For mental health practitioners and novice researchers, the multiple or even mixed features of gamers with problematic gaming behaviors could be difficult to identify. Therefore, the current study is based on a longitudinal dataset [32,33] to select a small sample of college gamers who were diagnosed as having problems in gaming. We focused on exploring their motives, lifestyles, and behaviors to assess how they constantly switched between the virtual and the real worlds, in a similar way to the migration decision.

Thus, this study aimed to identify gamers who may have high risks of gaming disorder from an understanding of their college life experiences. In study 1, a semi-structured interview and a two-cycle content analysis were conducted to form a code hierarchy for problematic gaming behaviors. In study 2, based on the diagnostic interview, two participants with both internet addiction and depression and two participants with internet addiction only were selected to develop a persona. The personas were then described based on the codes extracted from study 1. The purpose of developing personas is to form multiple portraits of the vivid characteristics of colleges students with problematic gaming behaviors, specifically regarding their lifestyles, gaming and Internet usage, psychological needs, and how and why they play online games.

## 2. Study 1

### 2.1. Method

#### 2.1.1. Participant Selection

Twenty participants (male = 17; female = 3), who were considered high-risk gamers, were selected from 757 college students in a longitudinal dataset [32,33] that investigated the sample over a span of 2.5 years. These high-risk gamers reported high problematic Internet use (Problematic Internet Use, PIU) [34] and high depression (Beck Depression Inventory, BDI-II) [35]. Their time spent on the Internet was over 4 h per day or above 6 days a week.

#### 2.1.2. Procedure

First, three psychiatrists conducted diagnostic interviews based on the criteria of the DC-IA-A [36] and depression (DSM-IV-TR) [37]. Semi-structured face-to-face interviews were then conducted with 20 college students identified with problematic gaming behaviour. The interview guidelines are listed in Table 1. The interview data were digitally recorded and transcribed verbatim.

Informed consent was obtained before diagnosis and interviews. This research was approved by the IRB (CCH-IRB-No.: 111219). The interviewees received incentives of NTD500 (USD16) for the interviews.

#### 2.1.3. Content Analysis

The interview data were transcribed verbatim and analyzed using N-Vivo 9.0 by the authors. The first cycle content analysis is exploratory and aimed to assist the readers to hear what the authors heard from our high-risk gamers and the “code” is the results of the first-cycle coding [38]. To extract meaningful codes, we were guided by three gaming motives (escapism, advancement, and socializing) and features of problematic gaming. The second cycle content analysis is explanatory and was guided by the PPM model. The second cycle produced three hierarchical concepts with the terms “category,” “theme,” and “topic” as suggested by Saldana [39].

The consistency of coding under the five themes was tested for the consensus of content analysis with Kappa Coefficients (Kappa Coefficients) [40] and results were satisfactory (advancement = 0.955, socializing = 0.958, escapism = 0.916, the core of real life = 0.910, symptoms of gaming addiction = 0.964), which demonstrated almost perfect (0.81–1) consistency between two coders [41].

### 2.2. Results

Table 2 shows the diagnoses of Internet addiction and depression, as well as the characteristics (i.e., gender and types of games involved) of these high-risk gamers. Among these 20 gamers, there were 17 male gamers (85%). Fourteen (70%) gamers played Massively Multiplayer Online Role-playing Games (MMORPG) [34,42], 12 (60%) gamers played Real-time Strategy Games [43], five (25%) gamers played Multiplayer Online Battle Arena (MOBA) [44], and one gamer played an Action Role-Playing Game (A-RPG) [45]. Among 20 high-risk gamers, 11 (55%) were diagnosed with Internet addiction who were identified as the gamers with problematic gaming behavior; among them four were suffering from both mental health problems from Internet addiction and depression. Nine gamers (45%) had no diagnosis of Internet addiction.

The MMORPG is an online role-playing video game where many people participate simultaneously. While playing MMORPG, the player assumes the role of a fantasy avatar in-game (also known as a “character”) and is responsible for his/her avatar’s actions and interacts with other players in-game [46,47]. Gamers “train” their avatars through accomplishing various missions every day. Due to its features of narrative, the game continues to exist in real time and “space” on the Internet even when a gamer logs out of the game or just takes a break [47]. The A-RPG is a form of role-playing game where the participants select characters that emphasize attacking. Compared with MMORPG, ARPG features single-player attack and promotion [48]. The MOBA is a subgenre of strategy video games where two teams of players compete against each other on a predefined battlefield [49]. Much of the strategy revolves around individual character development and cooperative team play in combat [49,50]. The RTS is also a subgenre of strategy video game. It allows all gamers to play in real time simultaneously [51]. Similar to a game of chess, the players usually fight against another human opponent in RTS [52].

#### 2.2.1. Results of Content Analysis

Following Saldana’s [38] content analysis guideline to select meaningful paragraphs from the interview transcriptions, we conducted initial coding. The content analysis was composed of two cycles. In the first, a total of 398 idea units and 26 codes were extracted (Appendix A). In the second, 11 sub-categories and five categories were formed. Two themes and one topic emerged.

Table 3 demonstrates the results of the content analysis. Five categories and 11 subcategories were found. These five categories formed two themes (T1 to T2): Push–Pull effects (T1) and Mooring effects (T2). Finally, the two themes formed the topic, “High-Risk Gamer’s Push–pull–mooring.” The first three categories (C1 to C3), Advancement and achievement (C1), Socializing (C2), and Escapism (C3) are descriptions of the urges that gamers actively seek in gaming. These three urges formed the “push-pull effect (T1).” The pull effect, shown in advancement and socializing in-game, refers to the great pleasure experienced by gamers from immersing themselves in a virtual world, and often to the extent of displaying addictive symptoms. The push effect, shown in escapism, is to describe gamers’ involvement in games because of their distress, pressure, and boredom in life. The fourth and fifth categories, shift focus to real life (C4) and Symptoms of addiction (C5) are descriptions regarding the back-and-forth process and consequences of persistent gaming. These two categories formed the mooring effect that refers to the gamers staying or lingering in online games (the virtual world), particularly the key factors of gamers staying in the game and not switching to the real world.

The coding principle for interview paragraphs is explained through the following example. I-1_C1_02 refers to the protocol of the participant I-1; the paragraph is coded to Category C1 (Advancement and achievement) and is shown as the participant’s second paragraph (02).

#### 2.2.2. Push–Pull Effect

##### Advancement (C1)

Games are designed with various achievement feedbacks (SC1.1). These feedbacks attract gamers to continuously learn and practice to upgrade game skills and levels (SC1.2). This repeated process is fascinating for gamers (SC1.1). There are explicit game rules and provided the gamers follow the rules, they can gain fixed or random rewards (SC1.2). Gamers are willing to invest more time and money or join a team to obtain honor and rewards (SC1.1). Games are so attractive (SC1.1) that college gamers may become immersed in thinking about games constantly, even in their classes (SC1.2). They make gaming a priority in daily activities; when they are free, they will access games immediately (SC1.2);

##### Socializing (C2)

Socializing is a crucial element in gaming. Some gamers play games for socializing, some gamers socialize to win the games, and some gamers enter the games because of loneliness and would like to look for friends (SC2.1). To communicate smoothly and make the game process more fun, the gamers would use voice devices or play games face-to-face (in Internet cafés or dorms) (SC2.2). Usually, gamers play games with familiar people, but sometimes they involve both familiar people and strangers (SC2.1). In addition to game platforms, other social network medias are used for communication and information exchange. Sometimes, gamers share common topics other than gaming experiences. They then establish heartfelt friendships and often chat with each other (SC2.2);

##### Escapism (C3)

The gamers enter the game world to escape temporarily from reality, especially when they perceive significant stress (e.g., when taking exams or feeling loneliness), have nothing to do, or feel vacant and uncertain (SC3.1). Games are designed to show explicit regulations and fixed rewards, which may offer gamers a sense of security and lessen uncertainty. When gamers have greater pressure, they may spend more time on games for escapist reasons when they should pay attention to dealing with stressors. For instance, when college gamers have an upcoming exam instead of focusing on that, they may spend more time on games to relieve stress (SC3.2);

#### 2.2.3. Mooring Effect

##### Shift Focus to Real Life (C4)

Gamers with problematic gaming behavior may sometimes have self-awareness of the adverse effects of gaming and that they should be away from their games. To return to real life and avoiding gaming, gamers may try several strategies. For instance, they attempt to reduce their time spent on the Internet and find part-time jobs or join school associations (SC 4.1); however, the sustainability and the effect of this are poor. Frustration could happen when gamers encounter obstacles in game level upgrades; they may reduce their gaming time or change to play other games (SC4.2). The mooring effect could be observed in this process of shifting from gaming to real life;

##### Symptoms of Addiction (C5)

Gamers may think they can only play one game and then stop, but the truth is they cannot stop and play dozens of games. They experience the feeling of being out of control from their failure to reduce gaming time (SC5.1). Adverse events in real life result from gaming including skipping class, failure in exams, and dropping out of school (SC5.2). Many gamers with problematic gaming behavior feel they have Internet addiction and should make some changes to be away from their games (SC5.3). They perceive their problematic gaming behaviors, but they fail to change. This makes them frustrated through repeatedly pulling themselves out of and then reentering the game.

## 3. Study 2

### 3.1. Method

#### 3.1.1. Participant Selection

Based on study 1, four gamers with problematic gaming behavior who had the richest data were identified from the 20 problematic college gamers (Table 3) and were chosen as the personas of this study. They were diagnosed though diagnostic interviews by psychiatrists. Two of those diagnosed were Internet addicted and other two were both Internet addicted and depressive (Table 2). Regarding the selection of prototypical gamers as the personas, we formed two rules. The first was to select cases with the richest information (codes extracted from study 1) and the highest contrast, in terms of gaming motives and consequences, from the previous ones selected. The second was to select cases based on the results of psychiatrists’ diagnostic interviews of Internet addiction by DC-IA-A [36].

#### 3.1.2. Persona Developing Process

The persona approach has been widely used in marketing or product design [53]; this process usually aims to better understand the characteristics of the target population (product users). In this study, the aim was to build personas that could represent various prototypes of college gamers with problematic gaming. Based on the two-cycle content analysis and the resulting code hierarchy (Table 3) in study 1, the persona stories of four selected gamers are presented in the following. The detailed codes for the four personas are provided in Appendix A. Specifically, two of the four gamers were not only diagnosed with Internet addiction, but were also diagnosed with depression by the psychiatrists based on the criteria of DSM-IV-R (Table 2). The other two gamers did not have depression diagnoses but were diagnosed as having Internet addiction.

### 3.2. Results

Presented here are the results of the qualitative, person-centered, first-person stories (personas) to describe the history of four cases on their gaming motives, dynamic processes, and life courses during college.

#### 3.2.1. The Four Gamers’ Persona Stories

##### The Persona I-1: Male Heavy Gamer Who Pursues Advancement and Socializing

Figure 1 shows the interview summary of the narrative about the first persona, I-1 (Internet addiction; non-depression, see Table 2). He showed a high involvement in gaming and a high level in all three motives (advancement, socializing, and escapism). He preferred to play the most popular MMORPG game and DotA real-time strategy game. I-1 is passionate about pursuing achievements (pull effect, advancement) in the game, enjoys being with other gamers, and easily becomes teammates with them (pull effect, socializing). He has good social ability and likes to play games with other boys side by side (pull effect, socializing). Due to his focus on online games, his academic achievement fell from his freshman to junior years (push effect, escapism). Across three years, he spent a lot of time playing games with roommates living in the same dorm (mooring in the virtual world). After his junior year, he moved out from the college dormitory and could not play games closely with his peers.

He gradually admits that hard work in a game is meaningless in real life and the gap between himself in the real world and himself in the virtual world had become insurmountable. In his senior year, I-1 decided to take the civil service examinations for the career building path. He realized that he is prone to losing self-control when gaming, so he began to take action to control the time spent playing games and surfing the Internet.

##### Persona I-2: Female Gamer Who Pursues Gaming Advancement and Escapes Boring Life

Figure 2 shows the interview summary of the narrative about the second persona, I-2 (Internet addiction; non-depression, see Table 2) who was selected because her persona was significantly different from the first persona. From her freshman to her senior year, I-2 focused on gaming. She pursued advancement in gaming (pull effect for advancement). To play games with her partners (pull effect for socializing), she spends a lot of time practicing game skills and earning points (pull effect for advancement) to keep up with the in-game levels for friends (pull effect for socializing). In real life, she cannot find the focus for life and has few friends (push effect for socializing). Although she can handle college studies roughly, she feels very mixed-up about life (push effect for escapism). I-2 does not join any student club and works part-time very briefly during college (push effect for escapism). In addition to playing games, I-2 also spends a lot of time reading online novels and animations (mooring in the virtual world). She feels a sense of emptiness about spending a long time online, but she cannot cope without the Internet.

##### Persona II-1: Male Immersed in Gaming to Escape the Loss of Love

Figure 3 shows the interview summary of the narrative about the third persona, II-1 (Internet addiction; depression, see Table 2) who started playing online games in freshman year with roommates (pull effect for socializing). He was immersed in the feeling of victory in gaming (pull effect for advancement). When he had important personal tasks (such as attending a student club) it was easy for him to leave the virtual world. When he unloaded his responsibilities in real life, he would be immersed in gaming (push effect, escapism). He refused to take the responsibility for his studies and often skipped classes. In his junior year, II-1 felt depressed because of the problems in an intimate relationship (push effect, escapism). Afterwards, he then fell into gaming almost all day (mooring in the virtual world), and his studies failed. In his senior year, II-1 started feeling worried about his future, so he began to study hard and cut off gaming. He deliberately found a job to keep himself busy and reduced the chances of being overly immersed in gaming. However, he admits that he still spent as much time as he could in games.

##### Persona II-2: The Lonely Male Gamer Who Was Looking for Friendship in Gaming

Figure 4 shows the interview summary of the narrative about the fourth persona, II-2 (Internet addiction; depression, see Table 2) who has limited friends in real life (push effect, socializing) and the social interaction in gaming makes him feel less lonely (push effect, escapism). He started playing online games in junior high school and considered online games the greatest fun in the world (pull effect, advancement). Almost all of his friends are gamers (pull effect, socializing). In real life, he cannot find the focus of life. II-2 has a low sense of responsibility for his studies, he always skips classes, and failed in many classes (push effect, escapism). In his senior year, other friends in games seemed to set their rea- life goals and would leave the game. He remained in the virtual world and could only play games with strangers (mooring in the virtual world). He also tried to leave the game, but work and studies in real life were very frustrating and pushed him to continue playing games (push effect, escapism).

## 4. Discussion

Among 20 gamers, there were only three female students recruited in study 1, which is not surprising because the gender distribution of gamers with problematic behavior was dominantly male as shown in previous studies [54]. In study 1, 398 idea units and 26 codes were extracted first and then 11 subcategories and five categories were formed. These five categories further formed two themes (T1 to T2): Push–Pull effects (T1) and Mooring effects (T2). Finally, the two themes formed the topic, “High-Risk Gamer’s Push–pull–mooring.” In study 2, four personas were selected and the first-person narratives were used to describe their gaming histories and daily courses during their college years. Through the four persona profiles, we found that understanding gamers’ multiple gaming motives and push–pull–mooring structure might be one of the ways to help them improve their addiction problem.

As for the results of the personas, the life stories of four gamers were narrated through a first-person view. These were cases with high gaming motives (advancement, socialization, and escapism) and were gamers diagnosed as having an Internet addiction by psychiatrists. Personas 1 and 2 are two gamers with no depression diagnosis, who started playing games due to a pursuit of advancement and enjoyment, which are pull effects from the virtual world, and their lives being boring and goalless, which are push effects; thus, they moored into the virtual world across their college years. For them, finding the shift to focus to real life was the key to mooring out from the virtual world. In addition to diagnoses of Internet addiction, the other two gamers in stories 3 and 4 were also diagnosed with depression. They dived into games because of affective problems of depression or loneliness that could be push effects from the real world, exerting force towards the virtual world. In contrast with the first and second personas, gaming might be an indispensable coping strategy for their depression and loneliness. It is suggested what we can help gamers with other ways to cope with depression or loneliness and to establish a focus on their real life.

### 4.1. The Characteristics of Gamers with Problematic Gaming Behavior: Gaming Motives

Three game motives, advancement/achievement, socializing, and escapism, were observed in every gamer in this study. In previous studies, these three motives are well recognized as related to problematic gaming [20,22,24]. Socializing and advancement are motives in most human activities, and may be termed positively, such as abilities [55,56]. Gaming is a spontaneous activity and the gamers could have feelings of achievement and earn friendship [57]. Therefore, we highly suspect that friendship and social interactions in games might not be too negative [15,20]; on the contrary, the gamers could enjoy the pleasurable process in gaining achievement and having social interactions [21,22,23,58].

However, the motive of escapism seems to be a much more negative motive for gamers who have gaming problems. For example, when college gamers show functional impairments, such as failing many exams and life duties, this could be a warning sign of further development of a gaming disorder. In this study, high risk gamers immersed in gaming to escape real-life pain, stress, emptiness, or boredom, showing that escapism is a powerful push force dragging them into the virtual world. This result is in accordance with the findings of the previous study [15]. Gamers with problematic gaming behavior reported that they usually had a sense of the possibility of their addiction; however, they struggled to change this status alone and they know outside assistance is necessary. In summary, long-term escapism seems to become the main motive in gaming and this could lead the gamers to higher gaming addiction risks [13].

When gamers cannot obtain socializing and advancement other than from gaming, they increasingly focus on gaming and escape to the virtual world. This could become a vicious cycle. This process represents an aspect of mooring in gaming. If the college gamers have weak social abilities in real life, and mostly obtain achievement from gaming, this could hasten the gamers’ mooring to gaming and staying online longer. While real life and virtual life are imbalanced, the push effect would be stronger and the mooring process would be prolonged, which may have negative consequences [58].

### 4.2. Gaming Seems to Be a Long-Term Coping Strategy and Consequence of Depression and Loneliness

For those gamers with problematic gaming behavior who had depression, gaming may be an important coping strategy for negative emotions. Previous studies [59,60] have shown that problematic gaming and depression are mutual comorbidities. In this study, there were four participants with both depression and Internet addiction diagnosed by psychiatric physicians among 11 Internet addiction participants from 20 gamers. Therefore, when dealing with gamers with problematic gaming behavior, a depression diagnostic interview is strongly suggested before forming an intervention plan. Once the emotional status of a gamer improves, it is likely that the gamer will not be as heavily dependent on gaming [61]. In other research about child gamers, a cross-lagged path analysis indicated a reciprocal relationship between the severity of the Internet gaming disorder and the level of depressive symptoms [60]. Therefore, depression should be monitored in gamers with problematic gaming behavior.

We also suggest that when gamers with problematic gaming behavior are depressed, medical treatment should be a priority and a major intervention over others. For clinical staff (such as counseling psychologists), parents, classmates, and friends of college gamers’ who have a problem, one effective strategy to help them avoid negative consequences is to help them find meaningful focuses in real life. It is important to note that some studies found that problematic gaming behavior may lead to gamers’ depression [60].

### 4.3. Preventing the Development of Problematic Gaming into Gaming Addiction

The finding of mooring effects revealed that when problematic college gamers shift their focus to real-life goals, for instance, joining student associations, having part-time jobs, engaging in previous hobbies (such as playing music instruments), or pursuing new and advanced goals (such as preparing to pursue a master’s degree), gaming is no longer the priority in life. The shifting process from the gaming world to the real world could start from a small, short-term goal, allowing a gamer with problematic gaming behavior to feel real-life accomplishment or achievement in a short time, such as walking for 30 min in the sunshine three times per week. Starting from a small goal, the gamers with problematic gaming behavior could gain a real sense of control again. From this sense of control, further expanded goals should be set, which may help them to find other tasks to achieve so that they may trigger the mooring process from the game world to the real world.

Currently, the Internet is broadly used for working and leisure connections with various electronic devices [62]. In terms of preventing problematic gaming use, keeping away from smart phones, tablet computers, and desktops is not a plausible solution. We suggest a program of Internet use literacy starting from childhood would be an effective alternative. An Internet use literacy program could be composed of activities such as asking young users: (1) to use Google Maps in searching for travelling destinations and planning routes; (2) to create and design a blog as a diary or journal; or (3) to play online games and introduce learning on how to find out tips for upgrading in games with parents [63]. These online activities could help children to use the Internet as a learning tool successfully, and reduce the possibility of problematic use of Internet [64]. Both in real life and virtual worlds, some advice could be proposed to prevent Internet addiction. For example, one could explicitly maintain a focus and pay attention during a task and enjoy the process; establish small achievements continuously in daily life; promote diverse interests and abilities in various activities; accumulate the Internet learning ability for self-learning; and increase self-identity and recognition by others from multiple sources.

## 5. Conclusions and Further Application

Three gaming motives, advancement and achievement, socializing, and escapism, were revealed from the content analysis of interview data which make a gamer persistently involved in gaming. The push–pull–mooring effects were constructed and based on this framework and the findings suggested that “the balance” between the virtual world and the real world is critical. When an imbalance tilts towards the virtual world, gaming becomes problematic. In addition, it is necessary to notice whether the gamers with problematic gaming behavior suffered from depression because medical intervention may be needed.

These findings could help mental health professionals to understand the mixed characteristics of gamers with problematic gaming behaviors. The effects of escapism and mooring processes may be used in planning interventions such as helping them to shift their life focus from gaming to authentic tasks in real life. There are two limitations in this study. The problematic college gamers in this study were purposely selected, therefore, the generalization and representability of the results to other samples in different ages should be considered. When applying the results to different game types, cultures and eras should be carefully considered.

## Figures and Tables

**Figure 1 ijerph-20-00798-f001:**
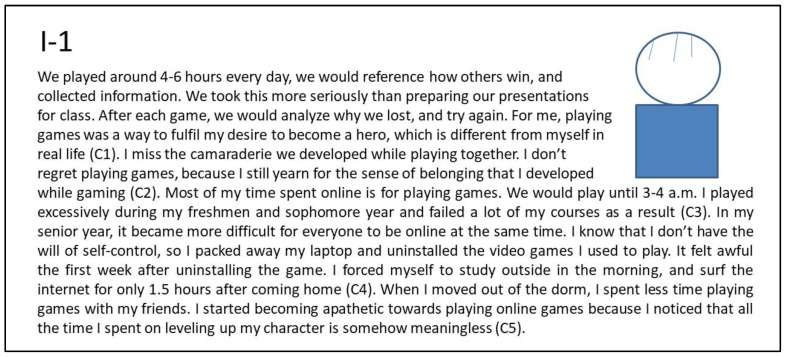
Person narrative about the first persona (I-1).

**Figure 2 ijerph-20-00798-f002:**
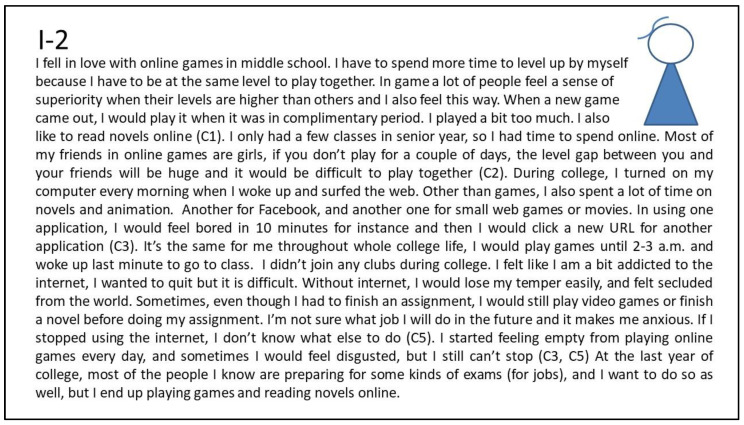
The person narrative about the second persona (I-2).

**Figure 3 ijerph-20-00798-f003:**
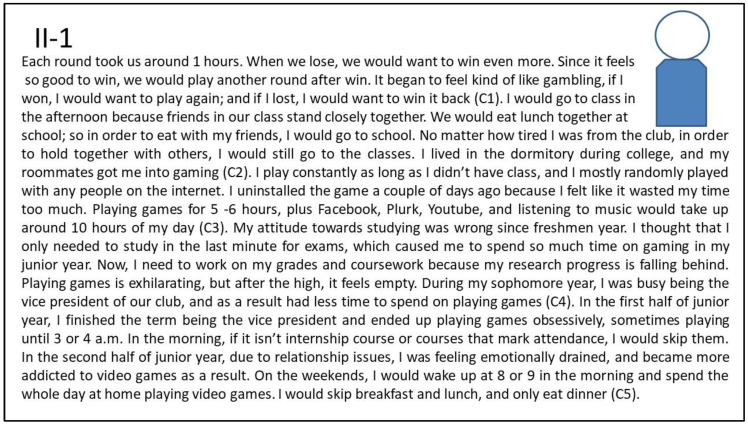
The person narrative about the third persona (II-1).

**Figure 4 ijerph-20-00798-f004:**
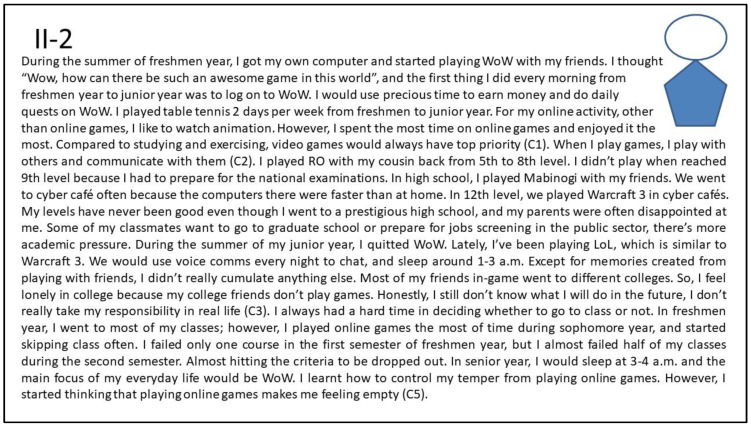
The person narrative about the fourth persona (II-2).

**Table 1 ijerph-20-00798-t001:** Interview guidelines.

1. Retrospective of Internet use in your life from freshman year to senior with five Ws: A. What kind of activities did you have online? What kind of devices did you use to get online? (e.g., computer, smart phone, iPad, etc.) B. When did you usually go online? How often? What was the longest duration? C. Who did you talk to online? Acquaintances or strangers? D. Where did you go online? E. How did you start the activities that you had online?
2. Further questions regarding Internet activities: A. What are those things that you do not like that much and are not that pleasant to do but spend a lot of time on in the Internet world? B. What are those things that are pleasant to do and you enjoy doing and spend a lot of time on in the Internet world? If the answers for the above two questions are not answered fully, three more questions will be added: 1. which year did you do these activities? 2. how long did it last? 3. when was there a change?
3. Another three supplementary questions: A. How do you spend your day? B. What do you think problematic Internet use is? Describe it with an example that impressed on you a lot. Do you think you have problematic Internet use? C. Do you shut down your computer on a regular basis? Or you will stay connected constantly? Will you respond to messages from anyone? At the end, in addition to what we talked about today, is there anything that you would like to add?

**Table 2 ijerph-20-00798-t002:** Description of characteristics of the participants.

Pseudonym	Gender	Game Type (Number of Game within a Type)	Internet Addiction Diagnosis ^a^	Depression Diagnosis ^b^
*** I-1**	M	MMORPG (1); RTS (1)	Y	N
*** I-2**	F	MMORPG (2)	Y	N
I-3	F	MMORPG (2)	Y	N
I-4	M	MMORPG (1); RTS (1); MOBA (1)	Y	N
I-5	M	MMORPG (1)	Y	N
I-6	M	RTS (2)	Y	N
I-7	M	RTS (1)	Y	N
*** II-1**	M	RTS (1)	Y	Y
*** II-2**	M	MMORPG (2); MOBA (1); RTS (1)	Y	Y
II-3	M	MMORPG (2)	Y	Y
II-4	M	MMORPG (1)	Y	Y
III-1	M	MMORPG (2); RTS (1)	N	Y
III-2	M	RTS (1); MOBA (1)	N	N
III-3	M	RTS (1); MOBA (1); MMORPG (1)	N	N
III-4	F	MOBA (1); MMORPG (2)	N	N
III-5	M	A-RPG (1); RTS (1); MMORPG (1)	N	N
III-6	M	RTS (1); MMORPG (1)	N	N
III-7	M	RTS (1)	N	N
III-8	M	MMORPG (2)	N	N
III-9	M	RTS (1)	N	N

Note. ^a^ DC-IA-A diagnostic criterion was used to define Internet addiction with gamers. ^b^ DSM-IV diagnostic criterion was used to define depression. * The gamer was chosen as the persona of study 2. (*n*) indicated how many games the gamers played within this type. MMORPG: massively multiplayer online role-playing game; A-RPG: action role-playing game; MOBA: multiplayer online battle arena game; RTS: real-time strategy game I: gamers with Internet addiction, but without depression II: gamers with Internet addiction and depression III: gamers without Internet addiction and depression (except III-1). The letter “Y” in the table means the pseudonym has internet addiction or depression diagnosis, which are been diagnosed by a psychiatrist, while the letter “N” means has not.

**Table 3 ijerph-20-00798-t003:** The second cycle content analysis and four personas.

Topic	Theme	Category	Sub-Category	I-1	I-2	II-1	II-2
Push–pull–mooring effects for Gamers with problematic gaming behavior	T1 Push–Pull effect	C1 Advancement and achievement	SC 1.1 Game level upgrade	*	*	*	*
			SC 1.2 Immerse in gaming	*	*	*	*
		C2 Socializing	SC 2.1 Socializing through online gaming	*	*	*	*
			SC 2.2 Socializing on other online platforms (Social Network sites)	*	*	*	
		C3 Escapism	SC 3.1 Antecedents of escapism	*	*	*	*
			SC 3.2 Behaviors of escapism	*	*	*	*
	T2 Mooring effect	C4 Shift focus to real life	SC 4.1 Reducing gaming time and Internet surfing for certain purposes	*	*	*	*
			SC 4.2 Avoiding using electronic devices	*	*	*	*
		C5 Symptoms of addiction	SC 5.1 Failure in self-control	*	*	*	*
			SC 5.2 Negative consequences	*	*	*	*
			SC 5.3 Self-awareness of addiction	*	*	*	*

Note. * There are some codes of the pseudonym (I-1, I-2, II-1, II-2) be included under the sub-category.

## Data Availability

Not applicable.

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
