# Peer review of "Developing Personas of Gamers with Problematic Gaming Behavior among College Students Based on Qualitative Data of Gaming Motives and Push–Pull–Mooring"

_ijerph, 2023, doi:10.3390/ijerph20010798_

Round 1
Reviewer 1 Report
I think it is a good manuscript that analyzes qualitative data on problematic gaming of college students and shows their reality and personas. If only a few things are supplemented, I evaluate that this manuscript is at a level that can be published in academic journals. Those things are as follows.
1. Please change “problematic college student gamers” to “problematic gamer among college students” in the title.
2. The introduction needs to be more focused on what has been done in this study. It is necessary to present the necessity of this study and the rationale of the research design in detail.
3. Before presenting the contents of Study 1 and Study 2, you should have explained in detail what Study 1 and 2 were conducted and why.
4. The conclusion needs to be summarized a bit more.
5. There are some incomplete references.
Author Response
Reviewer #1: Comments and Suggestions for Authors
I think it is a good manuscript that analyzes qualitative data on problematic gaming of college students and shows their reality and personas. If only a few things are supplemented, I evaluate that this manuscript is at a level that can be published in academic journals. Those things are as follows.
1-1 1. Please change “problematic college student gamers” to “problematic gamer among college students” in the title.
Response: Thanks for reviewer's reminder, we have rewritten the title as “Developing personas of gamers with problematic gaming behavior among college students based on qualitative data of gaming motives and push-pull-mooring”
1-2 2. The introduction needs to be more focused on what has been done in this study. It is necessary to present the necessity of this study and the rationale of the research design in detail.
Response: We appreciate the reviewer’s valuable suggestion and I have modified it, as you can read in lines 84-93.
lines 84-93
Thus, this study aimed to identify gamers who may have high risks of gaming dis-order from the understand of their college life experiences. In study 1, a semi-structured interview and a two-cycle content analysis were conducted to form a code hierarchy for problematic gaming behaviors. In study 2, based on the diagnostic interview, two participants with both internet addiction and depression and two participants with internet ad-diction only were selected to develop persona. The personas then were described based on the codes extracted from study 1. The purpose to develop persona is to form multiple portraits on the vivid characteristics of colleges students with problematic gaming behaviors, specifically regarding their lifestyles, gaming and Internet usage, psychological needs, how and why they play online games.
1-3 3. Before presenting the contents of Study 1 and Study 2, you should have explained in detail what Study 1 and 2 were conducted and why.
Response: We appreciate the reviewer’s valuable suggestion and I have modified it, as you can read in lines 84-93.
lines 84-93
Thus, this study aimed to identify gamers who may have high risks of gaming dis-order from the understand of their college life experiences. In study 1, a semi-structured interview and a two-cycle content analysis were conducted to form a code hierarchy for problematic gaming behaviors. In study 2, based on the diagnostic interview, two participants with both internet addiction and depression and two participants with internet ad-diction only were selected to develop persona. The personas then were described based on the codes extracted from study 1. The purpose to develop persona is to form multiple portraits on the vivid characteristics of colleges students with problematic gaming behaviors, specifically regarding their lifestyles, gaming and Internet usage, psychological needs, how and why they play online games.
1-4 4. The conclusion needs to be summarized a bit more.
Response: Thanks for reviewer's reminder, we have rewritten the conclusion.
lines 434-450
Three gaming motives, advancement and achievement, socializing, and escapism were revealed from the content analysis of interview data which make a gamer persistently involved in gaming. The push-pull-mooring effects were constructed and based on this framework, the findings suggested that “the balance” between the virtual world and real world is critical. When an imbalance tilts towards the virtue world, gaming become problematic. In addition, it is necessary to notice whether the gamers with problematic gaming behavior was suffered from depression because medical intervention may have needed.
These findings could help mental health professionals to understand the mixed characteristics of gamers with problematic gaming behaviors. The effects of escapism and mooring processes may be used in planning interventions such as helping them to shift their life focus from gaming to authentic tasks in real-life. There are two limitations in this study. The problematic college gamers in this study were purposely selected, therefore, the generalization and representability of the results to other sample in different ages should be carefully. When applying the results to different game types, cultures and eras should be carefully considered.
1-5 5. There are some incomplete references.
Response: Thanks for reviewer's reminder, we have added some references and rewritten all the article.

Reviewer 2 Report
This is an interesting paper with valuable insight into a much-needed area of research.
Abstract
The abstract introduces the content of the paper appropriately.
Introduction
The introduced theories and previous research results are in line with the content and direction of the paper.
Methods
Concerning participant selection, it would be necessary to introduce the longitudinal study as well to meet its characteristics.
Concerning the procedure, is it structured or semi-structured interview?
Results​
Results are introduced clear. ​The division of the chapters is logic, the introduction of the results of the analyses is correct. The tables and figures help the reader to understand the results.
Readers should be informed in advance that the paper introduces 2 research.
Line 122-131 would be better to be placed in the Methods/Participants subsection.
Figures are not real figures, they should be visualised only as quatations.
Discussion​
Discussion is correctly defined. The authors reflected well on previous findings and theories. This makes the paper coherent.
Conclusions
This section is proper as well.
Line 433-439: these findings are rather overall conclusions and characteristics og the gaming disorder, mot the result of the current studies
Limitations mentioned are correct. It should be also emphasised that results cannot be generalised to the population while the sample was not representative.
Overall, this paper is a valueable study which is worth publishing after minor modifications.
Author Response
For reviewer #2
Reviewer #2: Comments and Suggestions for Authors
This is an interesting paper with valuable insight into a much-needed area of research.
Abstract
The abstract introduces the content of the paper appropriately.
Introduction
The introduced theories and previous research results are in line with the content and direction of the paper.
2-1
Methods
Concerning participant selection, it would be necessary to introduce the longitudinal study as well to meet its characteristics.
Response: Thanks for reviewer's reminder, we have added references about original longitudinal dataset we mentioned in this article.
lines 97-99
Twenty participants (male = 17; female = 3), who were considered high-risk gamers, were selected from 757 college students in a longitudinal dataset [32] [33] that investigated the sample over a span of 2.5 years.
2-2 Concerning the procedure, is it structured or semi-structured interview?
Response: This study is a semi-structured interview. We wrote it at line 105-107.
lines 105-107
Semi-structured face-to-face interviews were then conducted with 20 college students identified with problematic gaming behaviour.
2-3 Results
Results are introduced clear. The division of the chapters is logic, the introduction of the results of the analyses is correct. The tables and figures help the reader to understand the results.
Readers should be informed in advance that the paper introduces 2 research.
Line 122-131 would be better to be placed in the Methods/Participants subsection.
Figures are not real figures, they should be visualized only as questions.
Response: Thank you for your suggestion, but we decided to fallow up the academic edit’s suggestion and remained the paragraph.
2-4
Conclusions
This section is proper as well.
Line 433-439: these findings are rather overall conclusions and characteristics on the gaming disorder, mot the result of the current studies
Response: Thanks for edit's reminder, we have rewritten the conclusion.
lines 434-450
Three gaming motives, advancement and achievement, socializing, and escapism were revealed from the content analysis of interview data which make a gamer persistently involved in gaming. The push-pull-mooring effects were constructed and based on this framework, the findings suggested that “the balance” between the virtual world and real world is critical. When an imbalance tilts towards the virtue world, gaming become problematic. In addition, it is necessary to notice whether the gamers with problematic gaming behavior was suffered from depression because medical intervention may have needed.
These findings could help mental health professionals to understand the mixed characteristics of gamers with problematic gaming behaviors. The effects of escapism and mooring processes may be used in planning interventions such as helping them to shift their life focus from gaming to authentic tasks in real-life. There are two limitations in this study. The problematic college gamers in this study were purposely selected, therefore, the generalization and representability of the results to other sample in different ages should be carefully. When applying the results to different game types, cultures and eras should be carefully considered.
2-5
Limitations mentioned are correct. It should be also emphasized that results cannot be generalized to the population while the sample was not representative.
Overall, this paper is a valuable study which is worth publishing after minor modifications.
Response: Thanks for edit's reminder, we have rewritten limitations.
lines 446-450
There are two limitations in this study. The problematic college gamers in this study were purposely selected, therefore, the generalization and representability of the results to other sample in different ages should be carefully. When applying the results to different game types, cultures and eras should be carefully considered.

Reviewer 3 Report
In their manuscript „Developing personas of problematic college student gamers based on qualitative data of gaming motives and push-pull mooring“ the authors report qualitative results from two studies. The aim of the study is to elucidate critical gaming motives for keep spending excessive time playing online games. The authors identify such critical “push effects” in terms of personas, indicating that “loss of purpose in life” and “desperate to escape from stress or boredom in the real world” play a crucial role in continued gaming. The methodological approach of the investigation is a strong point, particularly since sound qualitative analyses in this field are rare. The findings are interesting, especially since the lack of knowledge on maintaining factors for Gaming Disorder. Furthermore, the results are discussed properly and give important ideas for prevention and treatment strategies. Taken together, the study is of value, interest, and contains novel aspects that are helpful for many practical reasons. I have only some further suggestions that might be added in the discussion section.
(1) In study 1 the authors also included n=3 female high-risk gamers. Since female individuals are under-represented in research on Gaming Disorder, it would be interesting to know if you found gender-specific differences?
(2) The authors argue that depressive symptoms might be one important cause for keep playing games (in terms of self-medication). However, there are results from loingitudinal studies indicating that depressive symptoms might as well arise as a result of Gaming Disorder. I would like to encourage the authors also sharing some thoughts on depression being a consequence of gaming.
Author Response
For reviewer #3
Reviewer #3: Comments and Suggestions for Authors
In their manuscript „Developing personas of problematic college student gamers based on qualitative data of gaming motives and push-pull mooring“ the authors report qualitative results from two studies. The aim of the study is to elucidate critical gaming motives for keep spending excessive time playing online games. The authors identify such critical “push effects” in terms of personas, indicating that “loss of purpose in life” and “desperate to escape from stress or boredom in the real world” play a crucial role in continued gaming. The methodological approach of the investigation is a strong point, particularly since sound qualitative analyses in this field are rare. The findings are interesting, especially since the lack of knowledge on maintaining factors for Gaming Disorder. Furthermore, the results are discussed properly and give important ideas for prevention and treatment strategies. Taken together, the study is of value, interest, and contains novel aspects that are helpful for many practical reasons. I have only some further suggestions that might be added in the discussion section.
3-1 (1) In study 1 the authors also included n=3 female high-risk gamers. Since female individuals are under-represented in research on Gaming Disorder, it would be interesting to know if you found gender-specific differences?
Response: We appreciate the reviewer’s valuable suggestion and I have modified it.
lines 333-335
There were 20 gamers with only three female students recruited in study 1 which is not surprised, because the gender distribution of gamers with problematic behavior was dominantly male shown in the previous studies [54].
3-2 (2) The authors argue that depressive symptoms might be one important cause for keep playing games (in terms of self-medication). However, there are results from longitudinal studies indicating that depressive symptoms might as well arise as a result of Gaming Disorder. I would like to encourage the authors also sharing some thoughts on depression being a consequence of gaming.
Response: We appreciate the reviewer’s valuable suggestion and I have modified it.
lines 405-406
It is important to be noted that some studies found that problematic gaming behavior may lead to gamers’ depression [60].
